



# The effect of partial dissolution on sea-ice chemical transport: a combined model–observational study using poly- and perfluoroalkylated substances (PFAS)

Max Thomas[1], Briana Cate[1], Jack Garnett[2], Inga J. Smith[1], Martin Vancoppenolle[3], and Crispin Halsall[2]

[1]University of Otago, Department of Physics
[2]Lancaster Environment Centre, Lancaster University
[3]Sorbonne Université, Laboratoire d'Océanographie et du Climat (LOCEAN), Institut Pierre-Simon-Laplace (IPSL), CNRS/IRD/MNHN

**Correspondence:** Max Thomas (max.thomas@otago.ac.nz)

**Abstract.** We investigate the effect of partial dissolution on the transport of chemicals in sea ice. Physically plausible mechanisms are added to a brine convection model that decouple chemicals from convecting brine. The model is evaluated against a recent observational dataset where a suite of qualitatively similar chemicals (poly- and perfluoroalkylated substances, PFAS) with quantitatively different physico-chemical properties were frozen into growing sea ice. With no decoupling the model
performs poorly — failing to reproduce the measured concentrations of high chain-length PFAS. A decoupling scheme where PFAS are decoupled from salinity as a constant fraction, and a scheme where decoupling is proportional to the brine salinity, give better performance and bring the model into reasonable agreement with observations. A scheme where the decoupling is proportional to the internal sea-ice surface area performs poorly. All decoupling schemes capture a general enrichment of longer chained PFAS and can produce concentrations in the uppermost sea-ice layers above that of the underlying water con-
centration, as observed. Our results show that decoupling from convecting brine can enrich chemical concentrations in growing sea ice and can lead to bulk chemical concentrations greater than that of the liquid from which the sea ice is growing. Brine convection modelling is useful for predicting the dynamics of chemicals with more complex behaviour than sea salt, highlighting the potential of these modelling tools for a range of biogeochemical research.

## 1 Introduction

Sea ice is a complex, climatically important material, and provides a habitat for a range of micro-organisms (Vancoppenolle et al., 2013). As sea ice cools, the internal, liquid, brine becomes concentrated in solutes as more fresh, solid ice forms (e.g. Assur, 1958; Vancoppenolle et al., 2019). The increase in brine salinity raises the brine density (Maykut and Light, 1995; Cox and Weeks, 1988), which can drive brine convection (Notz and Worster, 2008) to supply nutrients (Fritsen et al., 1994) as well as harmful pollutants (Pućko et al., 2010a, b) to sea-ice communities. We can model salt dynamics and other conservative
species (chemical species that are completely dissolved and exhibit similar behaviour to salt) in growing sea ice with good accuracy and precision (Rees Jones and Worster, 2014; Griewank and Notz, 2013; Turner et al., 2013; Thomas et al., 2020). However, modelling chemicals with non-conservative behaviour (chemical species with behaviour that deviates from salt due





to physico-chemical or biological interactions within sea ice) has received less attention (e.g. Vancoppenolle et al., 2010; Zhou et al., 2013; Kotovitch et al., 2016). This problem has broad biogeochemical relevance, with gases such as $CO_2$ (Tison et al., 2002) and $CH_4$ (Zhou et al., 2014), macro-nutrients (Vancoppenolle et al., 2010), iron (Lannuzel et al., 2016), and stable water isotopes (Eicken, 1998; Smith et al., 2012) all having potential to behave non-conservatively with respect to salinity in sea ice.

A recent observational study gives us an opportunity to explore this problem is a systematic way. Garnett et al. (2021) measured the concentration profiles of a suite of ten poly- and perfluoroalkylated substances (PFAS) as they froze into laboratory grown sea ice. The measured PFAS share acidic functional groups (carboxylic and sulphonic acids) but exhibit differences in the carbon-chain lengths (ranging from 4 to 12) and their physico-chemical properties (see Table 1, where partitioning coefficients vary by several orders of magnitude). Intriguingly, Garnett et al. (2021) found that short-chain PFAS (carbon chain-lengths <8) behaved similarly to the bulk salinity (conservatively), while long-chain PFAS (carbon-chain lengths ≥8) were enriched relative to the bulk salinity. Further, long-chained PFAS tended to be even more enriched in the shallowest measured sea-ice layer, and in some cases had bulk concentrations greater than that of the underlying water. Given that brine convection is the dominant process redistributing solutes in growing sea ice (Notz and Worster, 2009), we hypothesise that the physico-chemical properties of the PFAS have partially decoupled them from the convecting brine, and that this decoupling has caused the observed deviations from conservative behavior.

Here, we test that hypothesis by using observations of PFAS (Section 2.1) to evaluate a brine convection model (Section 2.2.1). We ran three simulations that include plausible mechanisms of decoupling and tune each mechanism against the observations (Section 2.2.2). Comparing the model performance to the observations allows us to test our hypothesis (Section 3) and to provide insights into how brine convection parameterisations can be adapted to model non-conservative chemicals in sea ice (Section 4).

## 2 Methods

### 2.1 Observations

The PFAS observations were collected during two sea-ice freeze experiments performed using the Roland von Glasow Air-Sea-Ice Chamber (Thomas et al., 2021) where sea ice was grown from NaCl water. Results and methods are presented in full in Garnett et al. (2021). Two experiments were conducted (Freeze-1: air temperature $-35\,^{\circ}\mathrm{C}$, 3 days, 17 cm final sea ice thickness; Freeze-2: air temperature $-18\,^{\circ}\mathrm{C}$, 7 days, 26 cm final sea ice thickness), with samples extracted using the Cottier et al. (1999) method at the end of the growth phase, sectioned vertically, and measured for bulk NaCl and PFAS concentrations (Table 1). Following Garnett et al. (2021) we omit the lowest sea-ice layer from our analysis due to a sampling bias, and we exclude one PFAS (6:2 FTSA) as the reported concentrations are described as 'semi-quantitative'. We are left with measurements of bulk sea-ice concentration for NaCl and nine PFAS. The Freeze-1 and Freeze-2 depth profiles were subsampled into nine and eight vertical layers, respectively, giving 17 measured samples. We therefore have 17 measurements of bulk concentration, $c_{\mathrm{si}}$, for NaCl and 153 for PFAS. The concentration of the experiment water was measured three times when the system was fully liquid, and we use the mean and standard deviation of these as the concentration before freeze up, $c_{\mathrm{o}}$, and uncertainty





**Table 1.** PFAS used in this study. Initial water concentrations, $c_{\mathrm{o}}$, and analytical uncertainties, $\delta(c_{\mathrm{o}})$ and $\delta(c_{\mathrm{si}})$, were taken from Garnett et al. (2021). $N_{\mathrm{C}}$ is the carbon-chain length. $K_{\mathrm{OW}}$ is a measure of partitioning between a organic and aqueous phase and is a log exponent. The best tuning parameters for each decoupling method ($\alpha$, Section 2.2.2) are also given.

| PFAS | $c_{\mathrm{o}}$ / ng/L | $\delta(c_{\mathrm{o}})$ / ng/L | $\delta(c_{si})$/ ng/L | $N_{\mathrm{C}}$ | $K_{\mathrm{OW}}^{\dagger}$ | $\alpha_{\mathrm{B}}$ | $\alpha_{\mathrm{C}}$ / $10^{-5}$ kg.cm$^{-2}$ | $\alpha_{\mathrm{D}}$ / $10^{-3}$ kg.g$^{-1}$ |
|---|---|---|---|---|---|---|---|---|
| PFBS | 41.2 | 0.4 | 1.2 | 4 | 3.9 | 0.2 | 5.8 | 1.8 |
| PFPeA | 39.6 | 0.9 | 1.1 | 5 | 3.43 | 0.25 | 5.8 | 2.2 |
| PFHxA | 31.8 | 0.4 | 1.3 | 6 | 4.06 | 0.24 | 5.8 | 2.0 |
| PFHpA | 40.2 | 0.3 | 3.3 | 7 | 4.67 | 0.00 | 5.2 | 1.0 |
| PFOA | 38.6 | 0.2 | 0.7 | 8 | 5.30 | 0.63 | 7.6 | 6.4 |
| PFOS | 25.0 | 0.7 | 1.3 | 8 | 6.43 | 0.59 | 7.3 | 5.8 |
| PFNA | 24.0 | 0.5 | 0.5 | 9 | 5.92 | 0.76 | 8.8 | 8.3 |
| PFUnDA | 13.6 | 0.6 | 5.4 | 11 | 7.15 | 0.88 | 12 | 11 |
| PFDoDA | 10.7 | 0.8 | 6.6 | 12 | 7.77 | 0.81 | 9.7 | 9.3 |

$^{\dagger}$ Values from Smith et al. (2016).

on that concentration, $\delta(c_{\mathrm{o}})$, respectively. The uncertainty on a single measurement, $\delta(c_{\mathrm{si}})$, is taken to be the median standard deviation of the measurements of the fully liquid ocean because these have repeat samples. We normalise the concentration of NaCl and each PFAS to $c_{\mathrm{o}}$ and estimate the uncertainty on the normalised concentration by propagation of the measurement uncertainty. Normalising the concentrations allows us to compare the behavior of PFAS to each other and to NaCl. A full list of chemicals and selected key physico-chemical properties are given in Table 1.

## 2.2 Modelling

### 2.2.1 Model set-up

The model used was previously presented in Thomas et al. (2020), where brine dynamics parameterisations (Rees Jones and Worster, 2014; Griewank and Notz, 2013; Turner et al., 2013) based on Rayleigh number physics (Wells et al., 2011) were shown to perform well. The model has been used on several occasions to model tracers assumed to be perfectly dissolved in sea-ice brine (Thomas et al., 2020, 2021; Garnett et al., 2019), and has been used to investigate the sensitivity to imperfect dissolution for rhodamine (Thomas et al., 2020). We used the brine convection parameterisation presented by Griewank and Notz (2013) because it has a set of well evaluated tuning parameters, independent of our experimental system (Griewank and Notz, 2015), that allow us to test the robustness of our results to tuning (Section 4 and Supplementary Information). We tuned the parameterisation by varying the critical Rayleigh number, $Ra_{\mathrm{c}}$, and desalination strength, $\epsilon$, and evaluated the pairs using the bias and mean absolute deviation between the modelled and measured salinity profiles. We chose the pair $Ra_{\mathrm{c}} = 4.69$ and $\epsilon = 0.00438$ kg(m$^3$s)$^{-1}$ which gave a bias of less than 1 % and a weighted mean absolute deviation of 4 % when compared to





the 17 measurements of sea-ice bulk salinity. The model is initialised such that each sea-ice layer has the same bulk salinity and PFAS concentration as the underlying water, so the initial normalised concentration profile is identical for NaCl and each PFAS. Freeze-1 and Freeze-2 are modelled separately using temperature profiles and sea-ice thickness measured in each experiment (presented previously in Garnett et al. (2019)). The model has 50 layers and a timestep of 5 minutes. Chemicals are advected using

$$\frac{\partial c_{\mathrm{si}}}{\partial t} = -w \frac{\partial c_{\mathrm{br}}}{\partial z}, \tag{1}$$

where $w$ is the upwards brine velocity calculated using the Griewank and Notz (2015) parameterisation as described in Thomas et al. (2020) and $c_{\mathrm{br}}$ is the concentration of the chemical dissolved in brine. For salinity, $c_{\mathrm{br}}$ is a function of the local temperature (Weast, 1971). For PFAS,

$$c_{\mathrm{br}} = \frac{c_{\mathrm{si}}}{\phi} - c_{\mathrm{s}} \tag{2}$$

where $\phi$ is the brine fraction. Equation 2 expresses that the dissolved-in-brine PFAS pool is split into a mobile ($c_{\mathrm{br}}$) and a stationary phase ($c_{\mathrm{s}}$).

### 2.2.2 Methods of decoupling

For salt or other fully-mobile tracers $c_{\mathrm{s}}$ would be zero. For PFAS, $c_{\mathrm{s}}$ may be non-zero because of decoupling processes, where the term 'decoupling' here refers to a physical or chemical (or in other systems biological) process which removes PFAS from the free dissolved phase, thus preventing advection alongside brine in our model. We express the stationary phase as a fraction of the total brine concentration, such that

$$c_{\mathrm{s}} = \gamma c_{\mathrm{si}}/\phi, \tag{3}$$

where $\gamma$ is decoupling parameter; with the condition that $c_{\mathrm{s}} \leqslant c_{\mathrm{br}}$. As a first step (Method A), we model PFAS with no decoupling, such that

$$\gamma = 0. \tag{4}$$

Method A uses the same parameterisations as salt – so PFAS should behave identically to NaCl – and serves as a quality control for other model scenarios (Methods B to D) involving decoupling mechanisms.

Our simplest decoupling mechanism (Method B) alters the concentration of PFAS in brine as a constant fraction.

$$\gamma = \alpha_{\mathrm{B}}, \tag{5}$$

where $\alpha_{\mathrm{B}}$ is a free tuning parameter derived separately for each PFAS. Next (Method C), motivated by the surface active properties of PFAS, which tend to adsorb to surfaces (Grannas et al., 2013), the brine PFAS was fractionated to the stationary phase proportionately to the internal sea-ice surface area,

$$\gamma = \alpha_{\mathrm{C}} A_{\mathrm{si}}(\theta), \tag{6}$$





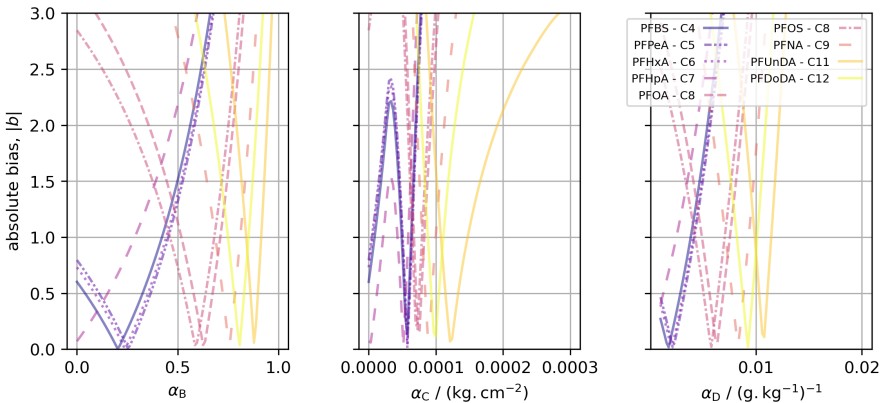

**Figure 1.** Absolute bias, $|b|$, between measurements and model with a range of tuning parameters, $\alpha$, for Methods B, C, and D.

where $\alpha_C$ is a free tuning parameter, and $A_{si}(\theta)$ is the internal sea-ice surface area as a function of the temperature, $\theta$, as given by Krembs et al. (2000) for columnar sea ice (see Thomas et al., 2020, Equation B1). Finally (Method D), we fractionate the brine PFAS concentration as a function of brine salinity, motivated by tendency of high ionic strength solutions to drive out solutes such as PFAS,

$$\gamma = \alpha_D S_{br}(\theta), \tag{7}$$

where $\alpha_D$ is a free tuning parameter.

While the brine salinity (Rees Jones and Worster, 2014; Weast, 1971) and internal surface area (Krembs et al., 2000) are both functions of temperature, the functional form is approximately opposite, with brine salinity decreasing and surface area increasing with temperature.

We get $\alpha$, for methods B to D and for each PFAS, by running the model using 100 values of $\alpha$ and selecting that which minimised the absolute bias between the measurements and co-located model layer at the end of the model run (Figure 1). For each method, short-chained PFAS tend to produce lower $\alpha$, and the two longest chained PFAS always give the highest $\alpha$ (Table 1, Figure 1).

## 3   Results

With no decoupling (Method A, using Equation 4) we see a systematic difference between the measured and modelled PFAS concentrations (Figure 2, a to c). NaCl concentrations are captured well (following salinity tuning in Section 2.2.1). All of the PFAS model lines are identical to that of NaCl but the observations shift to higher concentrations as the carbon-chain length moves from low (more blue) to high (more yellow), and some of the measured concentrations in the upper layer are above 1 (Figure 2, a and b). This shift is reflected in the correlation of measured and modelled concentrations, with lower chain lengths and NaCl clustered around the 1 to 1 line (perfect model behavior) but with longer chain lengths tending to lie above





the 1 to 1 line (Figure 2c). We use a weighted least squares regression of modelled vs measured concentration to quantify the model performance, using the inverse variance for each measurement as the weights. Good model behavior is taken to be a gradient, $k$, consistent with 1 (ideally within a standard error and at minimum within 95 % confidence intervals, 95%CI), and a

high coefficient of determination, $r^2$. The variability in the observations is barely captured ($r^2 = 0.15$), with the highest model concentrations less than 0.5 and the observations systematically above this for some of the long-chained PFAS. Though the gradient is consistent with 1 ($k = 0.93 \pm 0.16, 95\%\text{CI}[0.57, 1.28]$), the uncertainty is such that $k$ is poorly constrained. This regression is poor and motivates further model development.

Method B (using Equation 5) shifts the modelled PFAS concentrations higher (Figure 2, d to f). Model lines for the PFAS pro-

files can now be distinguished from each other and from NaCl but remain similar in shape. Longer chained PFAS have moved further from the NaCl profile and for PFUnDA and PFDoDA (chain lengths 11 and 12, respectively) concentrations rise above 1 near the upper interface. The regression is dramatically improved, with $r^2 = 0.66$ and $k = 0.91 \pm 0.05$ (95%CI[0.80,1.01]).

Surface area mediated decoupling (Method C, using Equation 6) also shifts the modelled PFAS concentrations higher (Figure 2, g to i). The profile shape is different to that of Method B. Moving from the lower interface up, we see a region of constant

PFAS concentration, then a decline in concentration, a region of near constant concentration, and finally an increase near the upper interface (see also Figure 3). To minimise $|b|$, $\alpha_C$ has become large enough to cause near total decoupling of PFAS near the lower interface where surface area is largest. This behavior is not observed, leading to an overestimation of PFAS in the lower sea ice and poor performance by the Method C regression ($r^2 = 0.29$, $k = 0.44 \pm 0.06$, 95%CI[0.33,0.55]).

Brine salinity mediated decoupling (Method D, using Equation 7), as with Methods B and C, shifts the modelled PFAS

concentrations higher (Figure 2, j to l). Moving upwards from the lower interface, concentration decreases sharply, then rises steadily until near the upper interface, then increases sharply towards the upper interface. The largest upper layer concentrations are predicted by Method D, reaching 1.25. The Method D regression ($r^2 = 0.67$, $k = 0.89 \pm 0.05$, 95%CI[0.79,0.99]) performs better than Methods A and C. Method D performs marginally worse than Method B with regards to gradient but captures slightly more of the variability.

We return now to our hypothesis that, in the experiments of Garnett et al. (2021), the physico-chemical properties of the PFAS decoupled them from convecting brine, resulting in deviations from behaviour conservative with salinity. The results for Method B are consistent with our hypothesis. The simple decoupling, implemented in a brine convection model, was sufficient to bring the modelled concentrations into quantitative agreement with the observations. Further, decoupling within brine and brine convection (particularly Method D) gave PFAS concentrations significantly above that of the underlying water, similar to

the observations.

## 4 Discussion

Methods C and D provide further insights into the mechanism of the decoupling. To explore the differences between the methods we take just one profile (PFOA for Freeze-1) and show the performance of each method alongside the observations (Figure 3). The change in bulk PFAS concentration is proportional to the vertical brine PFAS gradient when using the Griewank



**Figure 2.** Comparison of modelled and measured concentration for: Method A, perfectly dissolved chemicals (a, b, c); Method B, simple partitioning (d, e, f); Method C, surface area adsorption (g, h, i); and Method D, salting out (j, k, l). Depth profiles are shown for freeze 1 (a, d, g, j) and freeze 2 (b, e, h, k). Modelled against measured concentrations are shown in panels c, f, i, and l, alongside the best fit weighted least squares regression (black line, gradient $k$ with one standard error and coefficient of determination $r^2$ shown in legend) and the theoretical 1 to 1 line for perfect model behavior (dotted black). Concentration for the profiles is given on a log scale to highlight separation between the profiles.





and Notz (2013) scheme (Equation 1). Decoupling by surface area (Method C) was not a useful method in this case study as it could not simultaneously reproduce the shape and magnitude of the PFAS concentrations. Method C differs from Method B by including a dependence of the decoupling on internal sea-ice surface area, which increases non-linearly with temperature (Krembs et al., 2000). The profile shapes for Method C were qualitatively different to the observations and to Method B, and Method C performed quantitatively poorly, with the constant concentrations near the lower interface contributing most to

the poorer performance. Towards the base of the sea ice the PFAS are nearly completely decoupled due to the high surface area, so the vertical brine PFAS concentration gradient is near 0, causing little change in bulk PFAS concentration with brine convection. Bulk concentration therefore fixes at near the underlying water concentration until the sea ice grows, cools, and the surface area decreases (Figure 3). Method C cannot simulate the general increase in PFAS concentrations without unrealistically distorting their distribution. The results in this paper indicate that surface acting properties linked to internal surface area of the

sea ice was unlikely to control the behavior of PFAS in these experiments. Method D differs from Method B by including a dependence of the decoupling on the brine salinity, which decreases non-linearly with temperature. Higher brine salinities near the upper interface reduce the vertical brine PFAS concentrations most in the upper layers, and lead to preferential retention of PFAS higher in the sea ice. Method D performs similarly to Method B, though bulk concentrations are increased near the upper interface and reduced near the lower interface relative to Method B (Figure 3) – consistent with the vertical brine salinity

profile. Method D marginally fails one of our tests, where the gradient of the modelled concentrations regressed against the measurements is just inconsistent with 1 at 95 % confidence. However, given the improvement in model performance with Method D relative to Method A, and the similar performance between Methods B (which passed our tests) and D, our results do not rule out increased brine salinity driving PFAS out of solutions as an important decoupling mechanism in our study.

How general are our results? How robust is the brine convection tuning, and can our methods be applied to thicker sea

ice, other PFAS, and other chemicals? We tuned the model for this set of observations using NaCl measurements from the same samples as the PFAS measurements, which were extracted using the Cottier et al. (1999) method. While this method is the best sampling method we are aware of for young sea ice, retaining more brine than cores, it may still underestimate the sea-ice bulk salinity, causing the tuning parameters to give too much desalination (Thomas et al., 2021). Griewank and Notz (2015) present tuning parameters that perform well against observations when redistributing salt in simulations of Arctic sea

ice, and we test the robustness of our tuning by running a parallel set of model runs and analyses using the Griewank and Notz (2015) tuning parameters (Supplementary Information, Figures 1 and 2). Our conclusions are robust, with Method A performing poorly, Methods B and D improving model performance, and Method C performing poorly; and longer chained PFAS generally giving larger values of $\alpha$. The Griewank and Notz (2015) parameters perform worse than those derived for this study, but this is unsurprising and does not reflect poorly on the Griewank and Notz (2015) parameters, given we tuned to

salinity measurements made alongside the PFAS measurements.

Turning to the application of these results to thicker sea ice, we expect our results to generalise to thicker growing sea ice because brine convection will remain the dominant redistributor of solutes (Notz and Worster, 2009), with the caveat that the increased complexity in natural environments (transient periods of melt conditions, cracks in the sea ice, flooding of the surface) could significantly alter the redistribution of solutes beyond that driven by brine convection.


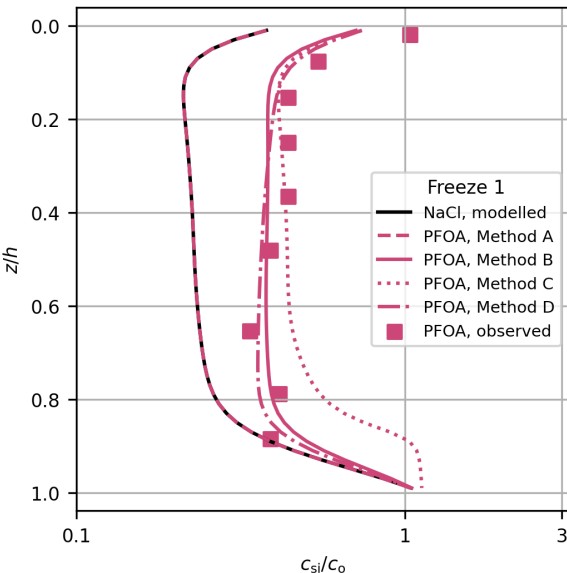

**Figure 3.** Comparison of modelled (lines) and measured (squares) PFOA concentration in Freeze 1 for each method. Modelled NaCl is also shown (black line).

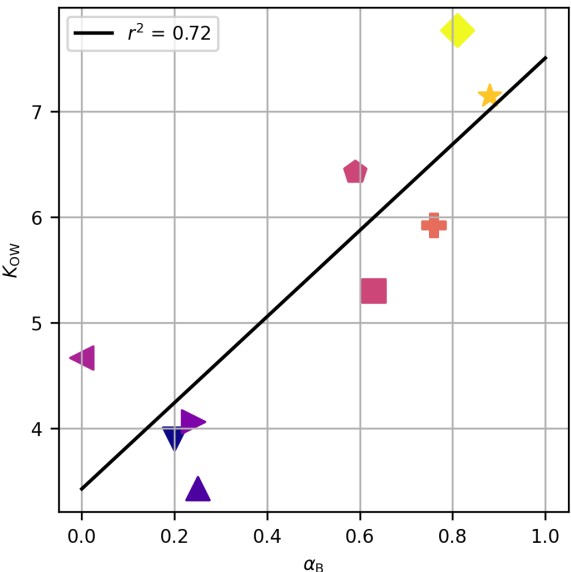

**Figure 4.** The octanol–water partitioning coefficient, $K_{OW}$, against the derived tuning parameter for Method B, $\alpha_B$, for each PFAS. Symbols are as in Figure 2. The black line shows the linear regression of $K_{OW}$ against $\alpha_B$, and the coefficient of determination is given in the legend.





For Method B (which performed satisfactorily) the tuning parameters derived for the PFAS, $\alpha_B$, are associated with the physico-chemical properties, as evidenced by the non-random ordering of $\alpha_B$ with chain length (Table 1, Figure 1). The partitioning coefficients presented ($K_{OW}$ in Table 1) tend to increase with chain length and are correlated with $\alpha_B$ (Figure 4, $r^2 = 0.72$, $p < 0.05$). This result builds confidence that our tuning parameters reflect real physico–chemical properties of the studied PFAS (though we acknowledge that organic–aqueous partitioning is of only indirect relevance to our experimental

system). In our experimental system, with close to neutral pH, all PFAS in this study exist in their anionic form as conjugate bases. We therefore have some confidence that, for PFAS not investigated here, taking the tuning parameter for the nearest chain length would be a useful approximation in natural seawater environments. For other chemicals, the different physico-chemical properties would necessitate re-tuning and possibly different functional forms for deriving $\gamma$, but the frameworks proposed for Method B (and possibly D) would be a useful starting point.

This study focused on PFAS, large organic pollutants, but the implications are more general. Many chemicals of interest behave differently to sea salt during sea-ice growth. Ammonia (Hoog et al., 2007), helium, and neon (Hood et al., 1998; Namiot and Bukhgalter, 1965) are partially incorporated within the ice matrix; isotopologues of water fractionate at ice–liquid interfaces (Lehmann and Siegenthaler, 1991); nutrients are consumed and remineralised (Vancoppenolle et al., 2010; Fritsen et al., 1994); gasses partition to bubbles and float upwards (Zhou et al., 2014; Kotovitch et al., 2016); and exopolymeric

substances stick to internal surfaces Krembs et al. (2011). We show here that quantitative agreement can be achieved for non-conservative chemicals in growing sea ice using brine convection modelling, highlighting the usefulness of these numerical tools to a range of biogeochimical sea-ice research areas.

## 5 Conclusions

A sea-ice brine convection model can reproduce observations of poly- and perfluoroalkylated substances (PFAS) freezing into

sea ice, providing a term is introduced that partially decouples PFAS from the moving brine. PFAS with longer carbon chains behaved less conservatively with respect to salinity and required larger decoupling parameters. Our results demonstrate that brine convection models are powerful tools beyond predicting sea-ice salinity. Complex, biogeochemical problems can benefit from the accurate, physically-based advection of brine predicted by current parameterisations.

*Code and data availability.* The full code, data, and software needed to reproduce this paper are available as a reproduction capsule at

https://doi.org/10.24433/CO.6237417.v1.

*Author contributions.* MT designed the research and wrote the manuscript. BC did the modelling runs and helped MT with the analysis. JG and CH provided expertise on the chemical properties. MV helped with the theoretical framework. IS won the funding and helped MT supervise BC. All authors reviewed the research and helped develop the manuscript.



*Competing interests.* The authors declare no competing interests.

*Acknowledgements.* MT's time on this research was partly supported by the Deep South National Science Challenge (Ministry of Business Innovation and Employment contract number C01X1412) and the Antarctic Science Platform's Project 4: Sea Ice and Carbon Cycle Feedbacks (University of Otago subcontract 19424 from Victoria University of Wellington's ASP P4 contract with Antarctica New Zealand through their Ministry of Business Innovation and Employment Strategic Science Investment Fund Programmes Investment Contract, contract number ANTA1801). The University of Otago Department of Physics 2021 Summer Research Scholarship scheme provided financial
support for BC's involvement in this research.



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
