# Peer review of "The effect of partial dissolution on sea-ice chemical transport: a combined model—observational study using poly- and perfluoroalkylated substances (PFAS)"

_The Cryosphere, 2023_

## Author Comment (AC1)

We thank Referee 1 for their comments and help improving the manuscript. This document replies to those comments point by point, with R1's comments in green, our responses in black, additions to the manuscript in blue, and deletions in red.

Max Thomas et al. determine if and how a 1D parametrization of brine convection can be expanded to reproduce the laboratory PFA measurements of a previous study (Garnett et al. 2021). The paper consists of roughly four parts. In part one, the authors propose four different methods (A,B,C,D). In part two, they tune the free parameters of methods B, C, and D to minimize the absolute bias. In part three, they compare the model against the observed PFA profiles. Finally, in part four, they discuss if their results would apply to longer sea-ice simulations and other chemicals of interest.

The paper's text is clear and the method is clearly formulated (with one small exception) and applied thoroughly. The main conclusions are clearly stated, relevant to the field, and well supported by the results. The plots are mostly legible and not misleading, and the code has been made fully available. The topic of the submitted manuscript fits The Cryosphere. Based on my experience, the quality of the draft is well above average.

However, I found the structure of the paper's second half confusing, and the paper is somewhat ambiguous about its scope. Moreover, the figures could be improved upon, and there are a few minor other issues to address. Accordingly, I recommend accepting the submitted paper, but with minor revisions.

**Minor comments in roughly descending importance**

- Scope. The introduction clearly states that the paper aims to determine if decoupling can explain the observed properties. However, the methods introduced and the results discussed go beyond that. I feel that one or two paragraphs are missing at the end of the introduction to describe the other main question of the paper, namely if the decoupling is linked to the surface area, brine salinity, or constant. Furthermore, here I feel that the expectations should be clearly stated. From the current draft, I am unsure which methods B, C, and D closest match the known theory.

  Several changes to the text now better describe the scope.

  We have added the following text to the introduction:

  > greater than that of the underlying water. It is plausible that
  > 15 PFAS were not perfectly dissolved in sea-ice brine during
  > these experiments. The extreme cold and high salinity of
  > brine has potential to drive solutes from solution, while the
  > surface active properties of PFAS could cause them to stick
  > to internal ice surfaces. Given that brine convection is the

  And to the final introduction paragraph:

25 Here, we test that hypothesis by using observations of PFAS (Section 2.1) to evaluate a brine convection model (Section 2.2.1). We ran three simulations that include plausible mechanisms of decoupling and tune each mechanism against the observations (Section 2.2.2). Comparing the
30 model performance to the observations allows us to test our hypothesis and identify the likely factors controlling PFAS dynamics in our experimental system (Section 3)and to. Finally, in Section 4 we provide insights into how brine convection parameterisations can be adapted to model non-
35 conservative chemicals in sea ice(Section 4).

We now describe more clearly how the good performance of Methods B and D are consistent with increased brine salinity driving PFAS out of solution:

similar dynamics. The gradient of the modelled concentra- 40
tions using Method D regressed against the measurements
is just inconsistent with 1 at 95 % confidence. However,
given the improvement in model performance with Method
D relative to Method A, and the similar performance be-
tween Methods B (which passed our testsproduced a gradient 45
consistent with 1) and D, our results do not rule out increased
brine salinity are consistent with increased $S_{br}$ driving PFAS
out of solutions as solution as being an important decou-
pling mechanism in our study. Both methods are able to

(The poor performance of Method C suggests that surface area is not the controlling factor leading to non-conservative behaviour. We state this in the original manuscript and have retained it.)

We also revised the conclusions:

A sea-ice brine convection model can reproduce observations
of poly- and perfluoroalkylated substances (PFAS) freezing
into sea ice, providing a term is introduced that partially de-
couples PFAS from the moving brine. PFAS with longer car-
30 bon chains behaved less conservatively were enriched with
respect to salinityand required larger decoupling parameters
., sometimes to levels above the underlying water, in
observations. Larger decoupling parameters were required to
simulate this behaviour. Decoupling methods with primary
35 dependence on the brine salinity performed well, and
outperformed a decoupling method that depended on the
internal sea-ice surface area. Our results demonstrate that
brine convection models are powerful tools beyond pre-
dicting sea-ice salinity. Complex, biogeochemical problems
40 can benefit from the accurate, physically-based advection of
brine predicted by current parameterisations.

• Structure. I missed the transitions between results, discussion, and conclusions on my first read. In my view, the tuning of the methods and the analysis of the resulting parameters are the first results. In the current draft, this is a single sentence at the end of 2.2, and is then revisited in Figure 4. Furthermore, I believe the results extend till line 173, and the discussion begins by discussing how general the results are. (I enjoyed the discussion along with the supplementary material.) From lines

129 to 173 I get lost between all the comparisons of B to C to D, and some things are repeated multiple times (e.g. lines 155-159). I recommend breaking down the results into more bite size chunks, answer a question and then move on to the next. One of the questions I would like to see answered is what it means that B and D are so similar. Is there no T dependence in reality? Or, is the data insufficient to distinguish?

On reflection, we retained Figure 1 and the text describing it in Section 2.2.2 (Methods of Decoupling). We feel that the parameter tuning is a technical detail, best placed there. We have revised the text to be clearer and have revised the figure caption to better introduce Figure 1 (please see new Figure 1 in response to the 3rd referee comment).

We have placed the beginning of the original discussion in the new results. The text has been reworked for clarity, and the similarity between methods B and D is now discussed (see below).

tests, where the (Figure 3). Despite this, Methods B and D are similar in terms of profile shape and quantitative performance. This similarity arises because Method B also includes a (implicit) dependence on $S_{br}$. The product $\gamma c_{si}/\phi$ (Equation 2) increases with $S_{br}$ (for both methods) because $\phi$ linearly scales with $S_{br}$. After tuning, both methods simulate similar dynamics. The gradient of the modelled concentrations using Method D regressed against the measurements is just inconsistent with 1 at 95 % confidence. However, given the improvement in model performance with Method D relative to Method A, and the similar performance between Methods B (which passed our tests produced a gradient consistent with 1) and D, our results do not rule out increased brine salinity are consistent with increased $S_{br}$ driving PFAS out of solutions as solution as being an important decoupling mechanism in our study. Both methods are able to bring the model into reasonable agreement with the available observations and further measurements would be needed to robustly distinguish Methods B and D.

• What is the absolute bias |b|? I assume it must be the absolute difference over the vertical sum of the modeled and measured concentrations. But in line 112, it says the difference between the measurements and the co-located model layers, which implies that |b| should only be zero when the model and obs match at all layers. Moreover, how is the absolute bias scaled? What does |b| = 1 mean? Why use the absolute value? Showing b instead could clearly show that the higher alpha is, the higher the total concentration is. It might also make the lines in Subfigure 1b less confusing.

We have explained this better in the text, and describe (and plot) the bias rather than the absolute bias, which is more intuitive (see next comment). Please note that residuals are signed so positive bias in one region can compensate for negative bias in another (i.e. the model does not have to match exactly at all layers to achieve b=0).

We choose $\alpha$, for methods B to D and for each PFAS, by running the model using 100 values of $\alpha$ and  picking the best performing $\alpha$. The performance was quantified using the bias, $b$, taken to be the sum of the residuals for each measurement relative to the co-located model  depth at the end of the  simulation. The best performing $\alpha$ gives $b$ closest to 0. For each method, short-chained PFAS tend to produce lower $\alpha$, and the two longest chained PFAS always give the highest $\alpha$ (Table 1, Figure 1).

- Figure 1 has many lines that are difficult to distinguish. The readability could be improved by increasing the plots' width to use the paper's full width. The yellow line is also difficult to see; a darker tone would be helpful. There are no subfigure labels (a,b,c),  and shifting the legend outside the area of the subfigure would also help. The current version, in which the legend blocks the lines' view and overlaps with the figure borders, is messy.

We have adjusted the colour scheme here (and for the rest of the manuscript). Rather than blue to yellow, the lines now go blue to red, which is good for colour blind readers and is clearer in general. Figure 1 now has a/b/c labels, the subplot spacing has been adjusted to maximise the space, and the figure spans more page width. The legend is now is panel b to avoid overlap with the lines. The lines are bolder. Another change is that we plot the bias, rather than the absolute bias, which is easier to interpret. The 0 bias line is added to highlight where the tuning parameters perform best. Some changes were made also in response to Referee 2. Please see the new figure below.

[Figure]

**Figure 1.**  Model tuning for each PFAS. The bias,  $b$, was calculated as the sum of the difference between measurements and co-located model  layers in three tuning  runs. Each panel shows results for  a different decoupling method: a) simple decoupling,  tuned using $\alpha_B$; b) surface area adsorption, tuned using $\alpha_C$; and  c) salinity mediated decoupling, tuned using $\alpha_D$. The best performing $\alpha$ give $b$ closest to 0 (black line).

- Figure 2 has too many lines and markers in too little space. This figure could be separated into two figures for profiles and scatterplots, but at least make full use of the paper width to make columns 1 and 2 twice as wide. This is now a minor detail, but I was initially confused by the axis choice for the right column. Since they share the same observation data, it makes more sense that the observation data be the x-axis—shared data on the shared axis. For example, one could easily compare where the 2.5 measured C12 is in each plot.

We prefer to keep the scatter plots in this figure, which captures all the main results for the manuscript. To improve the readability, we have: adjusted the subplot spacing; adjusted the subplot grid so that the profiles are as wide as the scatter plots; adjusted the colour scheme (see above); reduced the marker size; and adjusted the limits of the axes. We will also use the full available paper width. With regards to the choice of x/y axis for the plots, we have retained our original choice because the dominant uncertainty should be on the y variable in a weighted least squares and it is more intuitive to show depth profiles with depth on the y axis. Please see the revised figure below.

[Figure]

**Figure 2.** Comparison of modelled and measured concentration for: Method A, perfectly dissolved chemicals (a, b, c); Method B, simple  decoupling (d, e, f); Method C, surface area adsorption (g, h, i); and Method D,  salinity mediated decoupling (j, k, l). Depth profiles are shown for freeze 1 (a, d, g, j) and freeze 2 (b, e, h, k). Modelled against measured concentrations are shown in panels c, f, i, and l, alongside the best fit weighted least squares (WLS) regression (black line, gradient $k$ with one standard error and coefficient of determination $r^2$ shown in legend) and the theoretical 1 to 1 line for perfect model behavior (dotted black). Concentration for the profiles is given on a log scale to highlight separation between the profiles.

- Lines 170 and 172 reference some tests that can be passed or failed. I have searched the submitted manuscript and find no clue what tests these are.

We have removed the mention of 'tests' and have instead said precisely what we mean:

similar dynamics. The gradient of the modelled concentra-    40
tions using Method D regressed against the measurements
is just inconsistent with 1 at 95 % confidence. However,

- "was not a useful method" line 155. "useful" is not a well-defined adjective in this context. I recommend stating that C is worse than B and D and better than A.

We have amended to say that C is worse than B and D. We have not said C performs better than A, as it is not obvious that it does in all respects, with the observations being more poorly represented near the lower interface for short chained PFAS than method A.

I am trying to understand why the authors chose the name method A instead of reference or control. It is not a flaw and does not need to change, but I did find it strange that the first "method of decoupling" is "none". Accordingly, there is alpha_B, alpha_C, and alpha_D, but no alpha_A, and so on.

We chose to label the experiments with no decoupling as Method A because this is the most obvious way to model these chemicals. Please note that we did model each chemical during Method A, and lines for each PFAS are plotted in the profiles, but they are exactly overlain by the NaCl. We revised the wording when introducing Method A.

Method A uses the same parameterisations as salt — so PFAS should behave identically to NaCl — models PFAS    30
using the same numerical scheme as salinity, is the obvious choice for modelling solute dynamics, and serves as a quality control for other model scenarios (Methods B to D) involving decoupling mechanisms because PFAS should behave identically to NaCl.    35

---

## Author Comment (AC2)

We thank Referee 2 for their comments and help improving the manuscript. This document replies to those comments point by point, with R2's comments in green, our responses in black, additions to the manuscript in blue, and deletions in red.

**Overall comments**

This paper investigates the transport and evolution of chemical species through sea ice by comparing lab experiments with theoretical modelling work. The paper shows that simply assuming that PFAS evolve in the same way as salinity does not match observations. Three possible alternative models are tested, with two seeming plausible representations of this process. The rationales for some of these models could have been explained more and there are important presentational areas for improvement. But I think that the paper is a very good contribution, containing novel results of interest to the Cryosphere readership. Thus, it should be published subject to the minor corrections.

**Minor (general) comments:**

The presentation of results could be improved, both the in terms of the figures and the text. I make detailed line-by-line comments below, but I think the heart of the issue is that the figures may be in a sub-optimal order. The following is only a suggestion but may improve the readability of the paper.

Figure 3 is by far the easiest to understand, so I suggest it should go first, possibly with a second panel showing an example with a short chained PFAS as a contrast.

Then figure 1 could go second (or even by moved an appendix/supplement, which might allow for the model fitting to be explained in more detail). At present, it is not clear why the absolute value of b is plotted (or even the precise definition of b, which should be more clearly stated). Plotting b would make the sign of the bias apparent. The figure is far too small with very many lines (the choice of which is not precisely stated in the caption). One solution might be having a single panel (larger) in the main text explaining the calibration of a single example in greater detail, then move all the other calibrations to a supplement.

Then figure 2 (which was good) would move next, and finally figure 4. The latter could potentially be expanded to plot more of the correlations implicit in table 1. E.g., it could have a separate panel with N_c as the independent variable and alpha_{B,C,D} as dependent variables, normalized appropriately.

The reordering of the figures may also help the writing by making the ideas easier to visualize.

We thank the referee for these constructive comments.

On reflection, we have retained the original ordering of figures. Our reasoning is that:
Figure 1 shows tuning of the methods. It must therefore be placed before the other figures, which show data from tuned model results.
Figure 2 shows the scatter plots, which are described before the profile shapes are discussed in detail (for which we used Figure 3).
However, we have made adjustments to the figures and text to address the concerns of the Referee, and these have improved the readability of the manuscript. Figure 3 is referenced earlier, alongside Figure 2, so that readers are introduced to this easy to understand figure simultaneously with Figure 2. Figure 3 has also been placed in the Result section, rather than the Discussion. Also, in response to referee 1, the profiles in Figure 2 have been made more

clear (pasted below).

[Figure]

**Figure 2.** Comparison of modelled and measured concentration for: Method A, perfectly dissolved chemicals (a, b, c); Method B, simple  decoupling (d, e, f); Method C, surface area adsorption (g, h, i); and Method D,  salinity mediated decoupling (j, k, l). Depth profiles are shown for freeze 1 (a, d, g, j) and freeze 2 (b, e, h, k). Modelled against measured concentrations are shown in panels c, f, i, and l, alongside the best fit weighted least squares (WLS) regression (black line, gradient $k$ with one standard error and coefficient of determination $r^2$ shown in legend) and the theoretical 1 to 1 line for perfect model behavior (dotted black). Concentration for the profiles is given on a log scale to highlight separation between the profiles.

We thank the referee for highlighting the room for improvement in Figure 1. Figure 1 has been adjusted to 1) show bias, rather than absolute bias, as suggested; 2) have bigger panels; 3) have bolder lines; and 4) show the 0 bias line where we identify the best tuning parameters. The revised figure is pasted below. With these changes, the figure is more clear, and we retain all three panels in the main text rather than moving two to the supplement. Some of these changes were made in response to Referee 1, and more information about these changes are available in our response to Referee 1.

[Figure]

**Figure 1.**  Model tuning for each PFAS. The bias,  b, was calculated as the sum of the difference between measurements and co-located model  layers in three tuning  runs. Each panel shows results for  a different decoupling method: a) simple decoupling,  tuned using $\alpha_B$; b) surface area adsorption, tuned using $\alpha_C$; and  c) salinity mediated decoupling, tuned using $\alpha_D$. The best performing α give b closest to 0 (black line).

With Figure 3 we aim at communicating the differences between the methods. For short chained PFAS those differences are less clear (see figure pasted below), so we have not added a panel showing the short chain results. Several cosmetic changes to Figure 2 (in response to the other review) have made the profiles clearer.

[Figure]

We investigated different combinations of alpha_B and alpha_D regressed against K_OW and N_C (3 extra regressions, on top of alpha_B vs K_OW as shown in Figure 4). They are pasted below.

[Figure]

We did not investigate alpha_C as Method C did not perform satisfactorily. We state in the revised manuscript that:

> indirect relevance to our experimental system). We also calculated this correlation for $\alpha_D$ (which also performed well) and for $\alpha_B$ and $\alpha_D$ against $N_C$ (not shown). The correlations are all significant at 95 % confidence and range from $r^2 = 0.70$ to $r^2 = 0.81$. In our experimental system,

The other structural writing issue is the overlap between sections 3 and section 4. I think more of the detail should go into section 3 and then section 4 should be more summative (rather than restating the detail in section 3).

We have moved the start of section 4 to section 3, so that the comparison of the different methods is in the Results, and the Discussion starts with several questions. Some changes were also made in response to Referee 1, and these are detailed in our response to Referee 1.

**Technical line-by-line comments:**

- L5: could mention direction of bias (although this is clear later in the abstract)

    We have clarified that there is an underestimation (see also below).

- L6: "as a constant fraction" a bit vague (fraction of what, especially when it is being contrasted with "proportional to the brine salinity" which also has a constant proportionality coefficient)

    We have clarified:

    > ice. With no decoupling the model performs poorly —  underestimating the measured concentrations of high chain-length PFAS. A decoupling scheme where PFAS are decoupled from salinity as a constant fraction of their brine concentration, and a scheme where decoupling is proportional to the brine salinity, give better performance and bring the model into reasonable agreement with observations. A scheme where the decoupling is proportional to the

- L7: consider putting the other poorly performing scheme earlier (before the good ones)

    We have retained the original order, which was based on scheme simplicity. While we agree that the story would be more linear if the schemes improved from A to D, we feel that Method B must come second as it is more simple than C and D.

- L34: this was interesting, it looked like none of the models would have been able to reproduce this?

    Actually the decoupling schemes can reproduce this effect. Method D, for example, gives PFUnDA concentrations of 1.25 near the surface. We mention this in the

manuscript already (including in the abstract and results), but have added a line to the conclusions regarding this point:

> A sea-ice brine convection model can reproduce observations of poly- and perfluoroalkylated substances (PFAS) freezing into sea ice, providing a term is introduced that partially decouples PFAS from the moving brine. PFAS with longer carbon chains  were enriched with respect to salinity , sometimes to levels above the underlying water, in observations. Larger decoupling parameters were required to simulate this behaviour. Decoupling methods with primary dependence on the brine salinity performed well, and outperformed a decoupling method that depended on the internal sea-ice surface area. Our results demonstrate that brine convection models are powerful tools beyond predicting sea-ice salinity. Complex, biogeochemical problems can benefit from the accurate, physically-based advection of brine predicted by current parameterisations.

- L50: will the sampling bias be the same for salt vs PFAS? If the idea that some of the PFAS is stuck on the solid, then you might think there is a greater sampling bias for salt vs PFAS. Might be worth discussing

  We have added a line to the discussion.

  > underestimate the sea-ice bulk salinity. A brine loss driven sampling bias may impact PFAS differently to salt. If PFAS were preferentially retained, the observed enrichment in the sea ice would overestimate the in situ value. Also, tuning to samples affected by this bias would cause the tuning parameters to give too much desalination (Thomas et al., 2021). Griewank and Notz (2015) present

- Table 1: could give a definition/formula for K_OW.

  We have added the formula to the table and updated the notation to $\log_{10}K_{OW}$ throughout the manuscript.

- A more general point is that the terms 'decoupling,' 'partitioning,' 'fractionation' are being used for the same type of process. It may be worth spelling this out explicitly in the introduction.

  We have searched for these terms in the manuscript and have revised their use to be consistent. 'Partitioning' now only refers to K_OW. 'Fraction(ation)' now refers either to a literal fraction, or to isotopic fractionation.

- Eq. (1): could explain that diffusion is assumed to be slow on these timescales

  Rather than specifically mentioning diffusion we have cited Notz and Worster (2009) in two instances, stating that brine convection is the dominant redistributor of solutes in growing sea ice.

- L80: I would write out this formula. I would also make it more explicitly clear that c_br=c_si/phi (or similar expression using S for salinity).

We have added the formula (see also next comment).

> For salt or other fully-mobile tracers $c_s$  is zero, reducing to $c_{br} = \frac{c_{si}}{\phi}$ which is normally used
> 15 (e.g. Cox and Weeks, 1988).

- In equation (2), the meaning of this formula is not very clear. The usual lever rule for bulk concentration is c_si=c_br*phi+c_s*(1-phi), but this does not appear to be equivalent to equation (2).

We have written Equation two as follows :

Csi / ϕ = Cbr + Cs,

Which highlights that Cs and Cbr are to be read at the same level. We have also clarified that when C_s=0, this equation reverts back to the usual formulation and have cited Cox & Weeks (1988) as an example. We have also revised the text slightly to highlight that we made a choice to split the brine chemical pool.

$$\frac{c_{si}}{\phi} = c_{br} = \frac{c_{si}}{\phi} - c + c_s \qquad (2)$$

> where $\phi$ is the brine fraction. Equation 2 expresses
> 10 the choice of splitting the dissolved-in-brine PFAS pool  into a mobile ($c_{br}$) and a stationary phase ($c_s$).

- Relating to equation (3), subject to the comments about equation (2), it is possible to combine equations (2) and (3) to relate c_s to c_br, in which case the proportionality constant would be called a partition coefficient.

We agree that equations 2 and 3 could be combined. We have used the form we present in the manuscript because we need to introduce the stationary phase. If we combined equations 2 and 3 the stationary phase would disappear. While this would be a simpler formulation, the description in text becomes awkward.

- L104-105: could add a reference for this claim. Are there any independent experimental estimates of the strength of this effect?

While we are not aware of modelled or measured Setschenow (salting out) constants by sodium chloride for PFAS, the aqueous solubilities of these chemicals will be reduced in seawater/brine relative to pure water based on basic physico-chemical considerations. We have added a reference to Freire et al. (2005) who show this effect for Hexafluorobenzene.

-  add "(WLS)" as the acronym appears in figure legend but is otherwise undefined.

  We have defined WLS in the figure legend and in the results text.

- L144: the differences are quite marginal

  Please see new text discussing the similarities between Methods B and D (which was requested specifically by Referee 1).

   (Figure 3). Despite this, Methods B and D are similar in terms of profile shape and quantitative performance. This similarity arises because Method B also includes a (implicit) dependence on $S_{br}$. The product $\gamma c_{si}/\phi$ (Equation 2) increases with $S_{br}$ (for both methods) because $\phi$ linearly scales with $S_{br}$. After tuning, both methods simulate similar dynamics. The gradient of the modelled concentra-

- L165: needs a paragraph break (but also see general comments on rearranging sections 3 and 4).

  We have added this paragraph break. We have moved some discussion text to the results.

- L207: presumably, how to handle the decoupling will vary between these distinct types of chemicals?

  We agree, but think this paragraph is not the correct place to mention it. Here, we are discussing potential applications for brine convection modelling (and have avoided specific mention of decoupling). In the previous paragraph we discuss the generality of our results. We have made a change there, removing mention of a different gamma, and rather leave scope for more significant changes relative to the framework we propose.

  be a useful approximation in natural seawater environments. For other chemicals, the different physico-chemical properties would necessitate re-tuning and possibly different functional forms for  $\gamma$decoupling, but the frameworks proposed  here would be a useful starting point.